# Towards Text-Line Segmentation of Historical Documents Using Graph Neural Networks

**Kartik Chincholikar**[1]**, Kaushik Gopalan**[1] **& Mihir Hasabnis**
[1]Centre for Inter-disciplinary Artificial Intelligence (CAI)
FLAME University
Pune, India
`{kartik.chincholikar, kaushik.gopalan}@flame.edu.in`
`mihir.hasabnis@flame.edu.in`

## Abstract

We present an initial investigation into a graph-based problem formulation for performing text-line segmentation of historical documents, by representing characters (or grapheme clusters) as the nodes, and with edges connecting characters to their previous and next characters on the text-line. This converts the image segmentation learning task into a binary edge classification learning task. This also enables training on large-scale synthetic data simulating complex layouts, enabling better robustness to Layout-level distribution shifts observed in historical documents. Furthermore, we introduce a benchmark dataset of 15 Sanskrit manuscripts with diverse layouts. We propose a method based on CRAFT and Graph Neural Networks (GNNs), which uses geometric priors of text-lines to perform competitively with leading approaches in zero-shot and few-shot experimental settings on the Sanskrit dataset introduced and the U-DIADS-TL dataset. The proposed method further demonstrates competitive accuracy and better consistency than leading methods Doc-UFCN and SeamFormer when evaluating robustness to distribution shifts over increasing data sizes (using intra-manuscript and inter-manuscript train–test data splits) on the Sanskrit dataset introduced and the DIVA-HisDB dataset. Finally, we demonstrate that the proposed method achieves strong performance in the downstream, goal-oriented evaluation of text recognized from the segmented text-lines. The dataset, training, and inference code is available at: `https://github.com/flame-cai/gnn-synthetic-layout-historical/tree/gram-submission`.

## 1 Introduction

Digitizing text from historical manuscripts allows researchers to effectively search through them, track shifts in word usage over time, count the frequency of certain ideas, and perform natural language processing, providing a deeper understanding of a period's intellectual culture in ways that would otherwise be impractical (Chandna et al., 2016).

While Modern multimodal Large Language Models (MLLMs) have made significant progress in performing OCR on printed and digital copies of documents with various layouts (Wei et al., 2025; Cui et al., 2025; Poznanski et al., 2025; Liao et al., 2023; Xu et al., 2020), their capability to transcribe text from complex and dense historical manuscripts remains inconsistent due to misaligned visual-textual reasoning and over-reliance on linguistic priors (Crosilla et al., 2025; He et al., 2025; Coquenet et al., 2023).

Hence, extracting machine-readable text from such complex and dense historical manuscript pages involves a traditional two-step process: first, segmenting individual text-line images from manuscript pages and second, transcribing those line images into machine-readable Unicode text.

Segmenting text-lines from historical manuscripts has its own set of challenges. Annotation of text-lines and post-correction of predicted text-lines can be time-consuming to perform at a large-scale. Furthermore, the domain of the target documents may exhibit large distributional shifts relative to the training data (Kiessling et al., 2024; Nikolaidou et al., 2022). Sources of distribution shift can be

grouped into two categories: **Layout-level distribution shifts** and **Appearance-level distribution shifts**. Layout-level distribution shifts can arise from high heterogeneity in page layouts—including marginalia, interlinear glosses, footnotes, and irregular, curved text-lines. Appearance-level distribution shifts occur due to physical degradation artifacts (e.g., ink bleed-through, fading, staining), nuisance factors (e.g., camera noise, image compression), and low-level appearance shifts (e.g., faded ink, paper or palm-leaf textures, uneven illumination, darkened scans, variation in scribal styles).

These distribution shifts pose a significant challenge for deep learning models to generalize in-distribution (intra-manuscript train-test data split) and on held-out manuscripts (inter-manuscript train-test data split). This challenge motivated the FEST Competition 2025 (Few-Shot Text-Line Segmentation) (Zottin et al., 2025), which aims to stimulate research into designing models that generalize under intra-manuscript train–test data splits, when trained on as few as 3 annotated page images. With this context, our work makes three primary contributions:

**1) Problem Formulation.** We formulate the task of segmenting text-line images from historical handwritten manuscripts in a graph-based manner - by modeling the script characters (or grapheme clusters) as nodes, with edges connecting each character in a text-line to their previous and next characters. This converts the image segmentation learning task to a binary edge classification learning task. Additionally, this enables training GNNs on large-scale synthetic data simulating complex page layouts, enabling better robustness to Layout-level distribution shifts, leaving only the character detection module to be fine-tuned to Appearance-level distribution shifts of the target manuscript if required.

**2) Benchmark Dataset.** We introduce a benchmark dataset of 15 Sanskrit manuscripts with diverse layouts, with text-lines annotated using both Graph Neural Network(GNN) friendly labels, and bounding polygon labels (in the industry standard PAGE-XML format).

**3) Comparative Evaluation of Proposed Method.** We propose a method based on Character-Region Awareness For Text detection (CRAFT) (Baek et al., 2019) and GNNs, which can be trained using purely synthetic data simulating complex layouts to perform competitively with leading approaches Doc-UFCN and SeamFormer in zero-shot, few-shot, distributional robustness, and downstream-OCR experimental settings.

## 2 Literature Review

If the layout of a target manuscript from which text-lines are to be extracted is known *a priori*, it is usually possible to tune existing top-down text-line segmentation methods such as Projection Profiles (Chamchong & Fung, 2012) or Seam Carving (Nguyen et al., 2022) to perform well on that specific manuscript. However, this tuning is not manuscript-agnostic and typically requires hyperparameter adjustment for different target manuscripts.

Other pioneering deep learning based methods take in a manuscript image as input, and identify the text-lines either by predicting bounding polygons or by predicting pixel level masks (Jindal & Ghosh, 2023; Fizaine et al., 2024; Boillet et al., 2021; Vadlamudi et al., 2023; Kiessling, 2020; Oliveira et al., 2018; Grüning et al., 2018; Renton et al., 2018). Deep learning is applied to extract informative features from images for end-to-end text-line segmentation, or it is used as an intermediary step, with subsequent algorithmic post-processing.

However, deep neural networks, which perform dense pixel-level predictions, are known to be not robust to distribution shifts of various kinds (Das, 2021; Aubreville et al., 2021; Agrawal et al., 2025; Hendrycks et al., 2021). To alleviate this, recent methods LineTR (Agrawal et al., 2025) and CurT (Kiessling, 2022) model text-lines as piece-wise line segments and cubic Bézier Curves, respectively. These approaches use deep learning to extract information from the manuscript images and use it *to predict the parameters* that define the piece-wise line segments or cubic Bézier curves. In doing so, they make effective use of inductive priors and the geometric structure of text-lines.

We also draw inspiration from older works (Garz et al., 2012; Pastor-Pellicer et al., 2015; Rabaev et al., 2013), which first detect the locations of the characters (historically called interest points) and then connect the characters on the same line together algorithmically using inductive priors about text-lines. However, while these pioneering works determined character locations (interest points)

| File | Description |
|------|-------------|
| `[pg_id]_inputs.txt` | $(x, y)$ coordinates of each character on the page |
| `[pg_id]_labels.txt` | Graph-based labels: all points belonging to the same text-line have the same label. |
| `[pg_id].xml` | Bounding polygon labels: PAGE-XML file containing bounding polygonal text-line annotations. |
| `[pg_id].jpg` | Heatmap visualization of character locations (CRAFT output). |
| `[pg_id].jpg` | Original scanned images: Not included due to copyright restrictions, but are publicly accessible. |

Table 1: Dataset files corresponding to each manuscript page.

algorithmically, we use the deep learning based detector CRAFT (Character Region Awareness for Text Detection) (Baek et al., 2019). CRAFT produces a heatmap of character regions using a U-Net–style architecture (Ronneberger et al., 2015). The works by Leow et al. (2023); Shtok et al. (2021); Phung et al. (2020); Chincholikar et al. (2025) also make use of CRAFT to downstream applications in a similar manner.

Finally, we draw inspiration from applications of Graph Neural Networks to the broad field of 2D image understanding (Senior et al., 2025), where GNNs learn from data to capture relationships between objects in the image. Objects could be characters in the subfield of scene text recognition (Zhang et al., 2023; Xu et al., 2022; Zhang et al., 2020), words or text-lines in the subfield of Document Understanding (Davis et al., 2021; 2019; Lee et al., 2021; Nourbakhsh et al., 2024; Zhang et al., 2021; 2022; Gollbo, 2023; Wang et al., 2022), or wedge vertices when recognizing cuneiform signs (Kriege et al., 2018). Similarly, in the subfields of image captioning (Dong et al., 2021; Yao et al., 2018) and Visual Question Answering (Li et al., 2019; Liang et al., 2021; Nuthalapati et al., 2021), feature embeddings of objects in images (extracted using computer vision methods like Faster RCNN) are considered as node features, which are enhanced by GNNs by allowing nodes to exchange information in a learnable way, capturing spatial and semantic relationships between objects.

To summarize, our approach uses U-Net-based character detection, geometric inductive priors of text-lines, and GNNs in a complementary manner to perform text-line segmentation.

## 3 DATASET AND SYNTHETIC DATA GENERATION

**Sanskrit Dataset.** We introduce a dataset consisting of **15 handwritten Sanskrit manuscripts**, adding up to **481 pages**. We observe both Layout-level variations as well as Appearance-level variations in the dataset pages, and hence we stratify the dataset pages by the manuscript they belong to and by their layout type. We consider pages with a single column of text as **Simple layouts (178 pages)**, and consider layouts with marginalia, interlinear commentary, and uneven text sections arranged vertically, horizontally, and along curved paths as **Complex Layouts (303 pages)**, as illustrated in Figure 3 in Appendix A. The dataset is labeled in both graph-based labels and bounding polygon labels in the form of standard PAGE-XML files, as elaborated in Table 1

**Synthetic Data Generation.** Representing the text context of a page as a graph enables us to generate large-scale synthetic data simulating complex layouts in the graph-based data format discussed previously. To do this, we implement a synthetic data pipeline that uses rejection sampling to mix diverse content types (main text, marginalia, interlinear glosses, page numbers). The pipeline supports multi-stage data augmentation by micro-jittering the character locations to simulate handwriting inconsistency, and by applying distortions (shear, curl, and linear creases). We also implement a "text-box splitting" strategy that randomly splits one text-box into two resultant text-boxes, and then applies a small shift and rotation to one of the text-boxes. This simulates severe structural ambiguity at the text-box level, implicitly training the GNN to resolve complex boundary decisions through data.

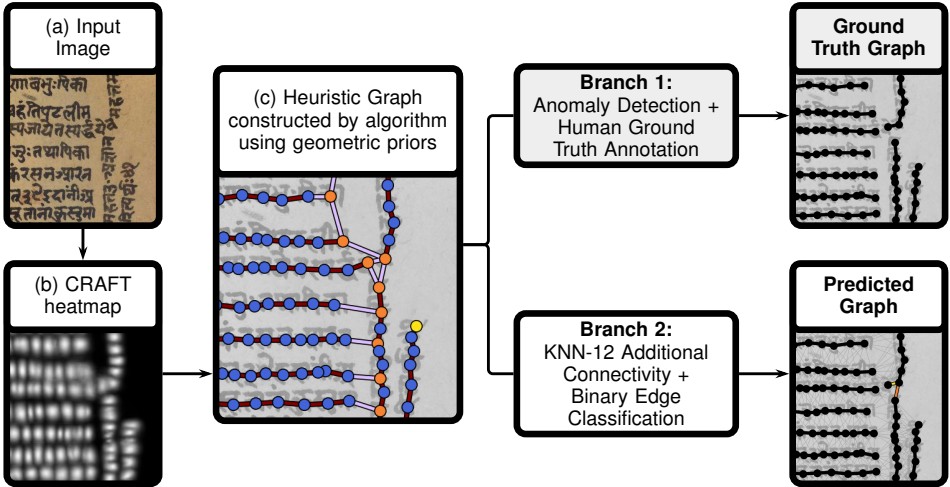

Figure 1: **Graph-based problem formulation and Pipeline for text-line segmentation.** An original manuscript image (a), which is converted into a heatmap using CRAFT (b). From this, a heuristic graph is constructed using an algorithm that incorporates geometric priors of text-lines (c). The colors of the nodes in (c) represent the degree of the node, while the colors of the edges represent the number of times that edge was chosen by a node (once or twice). These colors are *informative*, and are represented as *one-hot* node and edge features. At this step, the pipeline splits into two branches. Branch 1 performs crude anomaly detection, followed by human annotation to get ground truth labels. Branch 2 creates additional graph connectivity using K-Nearest Neighbours and then performs binary edge classification using GNN Layers to get the predicted graph.

## 4    METHOD

First, we use CRAFT (Baek et al., 2019) to locate the characters on a page, thus creating a set of nodes where each node's features are the X, Y coordinates of the character it represents. Next, we build a heuristic graph using an algorithm that incorporates the following geometric priors of text-lines:

(a) Each node (character) in a text-line can have at most two edges (connecting to previous and next characters), with the angle between the edges being $\sim\mathbf{180}°$

(b) The character spacing is less than the line spacing for most writing systems.

This algorithm acts on each node *locally* and is *invariant* to page translations, rotations, and the order in which it acts on the nodes. For each node, we first find the 10 nearest nodes (using Euclidean distance). We then form all possible pairs of these 10 nodes and filter only those pairs where the angle between the edges from the current node to each of the neighbours in the pair is $180° \pm 40°$. From these filtered pairs, we choose the pair with the shortest edge lengths, as the sum of edge lengths to the previous and next characters would be the shortest, assuming that adjacent characters in the same line are closer than nearby characters in the adjacent lines. This gives us the heuristic graph, as seen in Figure 1 (c), where the node colors denote the degree of the nodes, while edge colors denote the number of times an edge has been selected by a node. These node and edge colors produced by the algorithm are encoded as one-hot features alongside other geometric features.

At this step, the pipeline splits into two branches. In branch 1, as seen in Figure 1, we perform anomaly detection using the geometric features to filter out outlier triplets from the heuristic graph. This is achieved by applying DBSCAN (with parameters $\epsilon = 10$, min_samples $= 2$), a density-based clustering algorithm, to the geometric feature vectors derived from each potential connection. We identify the single largest cluster, assuming it represents the most common and correct connection pattern on the page. Only triplets belonging to this dominant cluster are considered valid connections; all others, including those in smaller clusters or marked as noise by DBSCAN, are treated as anomalies and discarded. For pages with complex layouts where the anomaly detection

Table 2: Summary of Node and Edge features. Features marked with a dagger ($^\dagger$) are the **discrete, one-hot features** obtained using the heuristic algorithm, explicitly making use of geometric priors of text-lines, whereas the unmarked features are **raw continuous geometric properties**.

| Level | Feature | Formula / Symbol | Description |
|-------|---------|------------------|-------------|
| **Node** | Norm. position | $(x_{\text{norm}}, y_{\text{norm}})$ | 2D coordinates normalized w.r.t. page size (maintaining aspect ratio) |
| | Node degree$^\dagger$ | $d_i$ | Number of edges connected to node (One-hot feature) |
| **Edge** | Rel. X disp. | $dx = x_t - x_s$ | Horizontal offset |
| | Rel. Y disp. | $dy = y_t - y_s$ | Vertical offset |
| | Euclidean dist. | $\sqrt{dx^2 + dy^2}$ | Distance between nodes |
| | Aspect ratio | $\frac{|dy|}{|dx|+10^{-6}}$ | Edge orientation measure |
| | Edge Overlap$^\dagger$ | $n$ | Number of times the edge was chosen by a node (One-hot feature) |

underperforms, a human annotator performs additional manual annotations by adding or deleting edges as required, using a semi-automatic layout annotation tool.

In branch 2, a Graph Neural Network is employed to replace the DBSCAN algorithm of branch 1 and perform anomaly detection in a learnable and more refined manner. We frame this problem as binary edge classification, where the GNN has to learn which edges to 1 (keep) or 0 (delete). To do this, we need to first ensure that the constructed graph edges are a superset of the true graph edges. As the edges of the heuristic graph constructed in the previous stage are not a superset of the true graph edges, we add additional graph connectivity by connecting each node to its closest 12 neighbouring nodes. These additional edges are assigned similar geometric features and a unique one-hot feature; however, the additional edges are not considered when deciding the degree of the nodes (which is a one-hot node feature). This additional connectivity also facilitates better communication between nearby nodes. This process completes the construction of the input graph $G = (V, E)$, with node, bi-directional edge, and graph features as documented in Table 2.

The graph is then passed through two sequential GNN layers, evolving the initial node features ($x_u$) into rich latent node representations ($h_u$). Finally, to perform binary edge classification on an edge, the edge feature, and the latent node representations of the nodes (which the edge connects) are concatenated and passed through a trainable Multi-Layer Perceptron (MLP) to get the logits.

Following the categorization by Bronstein et al. (2021), we investigate three types of GNN layer designs, representing increasing complexity in node-neighborhood interaction: a) Graph Convolutional Networks (GCNs, Kipf (2016)), b) Graph Attention Networks (GATs, Velickovic et al. (2017); Brody et al. (2021)), and c) Message Passing Neural Networks (MPNNs, Gilmer et al. (2017)). The hierarchy (GCN $\subseteq$ GAT $\subseteq$ MPNN) highlights the expressivity of the GNN layers, where GATs can be thought of as a specific instance of MPNNs, and GCN can be thought of as a specific instance of GATs. In this regard, it is of note that edge features are not used by GCNs and GATs in their layer-wise node representation update, and are only used in their final edge classifier. We also experiment with SplineCNN (Fey et al., 2018). In SplineCNN, a learnable, continuous kernel function maps the edge feature to a scaler, which decides the weight for node feature aggregation of the respective sender. Because this scaler is calculated dynamically based on the edge feature, and is not determined from the Adjacency Matrix A, SplineCNN can be considered as a specific instance of MPNNs. Additionally, we modify the SplineCNN architecture by forward-passing edge features through a learnable MLP as preprocessing, which reduces the edge feature dimensions, making them more suitable for SplineCNN.

As the number of "delete" edges vastly outnumber the "keep" edges, causing class imbalance, we use focal loss as the loss function, with $\alpha = 0.9$ and $\gamma = 2.0$. The optimizer used is Adam (Kingma, 2014), and all models are trained for 30 epochs, with a batch size of 4 and a learning rate of 0.001, with 6 epochs of learning rate warm-up. The early stopping is set to 15, with the validation metric

being an object-level metric which calculates how many text-line were correctly formed after the edge deletion.

To summarize, as seen in Figure 1, branch 1 can be used to semi-automatically annotate a dataset, and branch 2 is trained on this annotated data to teach the GNN to perform better than anomaly detection. Branch 2 can also be trained on purely synthetic data with diverse layouts.

## 5   EXPERIMENT SETUP

**Evaluation Metric.**   We compare the performance of the proposed graph-based method with leading text-line segmentation methods, Doc-UFCN (Boillet et al., 2021), which is a U-Net-based method, and SeamFormer (Vadlamudi et al., 2023), which is a Vision Transformer-based method. These methods frame text-line segmentation as a *dense pixel-level prediction task*, yielding bounding polygons that enclose the text-lines. In comparison, the proposed graph-based method splits the text-line segmentation task into two stages: 1) character segmentation and 2) binary edge classification. Binary edge classification aims to connect characters belonging to the same text-line together, which in turn labels each node (character) belonging to the same text-line with the same label. To ensure a fair comparison between graph-based predictions, with bounding polygon predictions, and to tackle label bias observed by Boillet et al. (2022), there is a need to perform label conversion from graph-based format to bounding polygon format as discussed in detail in Appendix B. To evaluate predictions in bounding polygon format, Boillet et al. (2022) recommend using the object-level metric Average Precision (AP@0.5) rather than pixel-level metrics such as Intersection over Union (IoU) and F1-score. Pixel-level metrics do not provide information about the number of correctly predicted, missed, or split objects and are more susceptible to label bias, causing misleading results (further discussed in Appendix B); therefore, we primarily report the results using the **AP@0.5 metric**.

**Zero Shot Learning.**   We define a zero-shot setting as when the proposed method is trained on synthetic data of 50K data points, with the pretrained character segmenter (CRAFT) being used as-is. We compare performance with state-of-the-art publicly available pre-trained models Doc-UFCN and SeamFormer, using all 481 pages of the Sanskrit Dataset introduced. Evaluations are reported separately for data-splits of pages with simple layouts and those with complex layouts. In addition to the Sanskrit dataset introduced, this experiment was also performed on the U-DIADS-TL (Zottin et al., 2025) dataset, with implementation details and results documented in Appendix D.

**Distributional Robustness.**   We evaluate the accuracy and consistency of the competing methods, with increasing data sizes, on both intra-manuscript and inter-manuscript train–test data splits. We define the *intra-manuscript train–test split* experimental setting as one in which the training, validation, and test sets each contain pages drawn from all manuscripts. We define the *inter-manuscript train–test split* experimental setting as one in which the training, validation, and test sets contain pages from different manuscripts, with no manuscript appearing in more than one split. Within each setting, we train competing methods on progressively larger training datasets to measure performance scaling, and repeat each experiment with 5 random seeds to generate 5 data splits. To ensure fair comparisons, all competing methods use identical data splits where the validation and test sets are fixed for each random seed. For each seed, only the size of the training set changes, and larger training sets are always supersets of the smaller ones. This guarantees that any performance differences for a method across training ratios are caused solely by the amount of training data, rather than by differences in the evaluation distribution. For training both Doc-UFCN and SeamFormer, we pay careful attention to the recommended configuration settings recommended by the authors (https://gitlab.com/teklia/dla/doc-ufcn, https://github.com/ihdia/seamformer), accessed on 12th November, 2025. All versions of the proposed method (SplineCNN, MPNN, GAT, GCN) are trained on real data (no synthetic data), with each data point augmented 20 times at the graph-based layout level. In addition to the Sanskrit dataset introduced, we also performed this experiment on the standard Latin dataset DivaHisDB (Simistira et al., 2016), with implementation details and results documented in Appendix C.

**Ablation Study.**   We conduct an ablation study to evaluate the dependency of the GNN architectures (GCN, GAT, MPNN, and SplineCNN) on the one-hot features obtained by the heuristic algorithm (node degree and edge overlap) and the raw continuous geometric features (normalized

2D coordinates, horizontal and vertical offsets, Euclidean distances, and aspect ratios) as seen in Table 2. Models were tested under four configurations: using full features, removing the one-hot features, and removing the raw continuous geometric features. Further details of this experiment are provided in Appendix E.

**Downstream Evaluation.**   We evaluate the downstream task of recognizing text content from historical page images using the page-level Character Error Rate (CER) as prescribed by Boillet et al. (2022) and detailed in Appendix F. Our experiments use two previously unseen manuscripts, one with a simple page layout (50 pages) and one with a complex page layout (15 pages). For text-line segmentation, we apply the pretrained competing methods Doc-UFCN, SeamFormer, and the proposed method (GNN-SplineCNN). For text recognition from segmented text-lines, we use (1) **Gemini-2.5-Flash**, by passing the predicted text-line coordinates as part of the prompt; (2) a **pretrained CNN–BiLSTM–CTC** recognizer; and (3) a **finetuned CNN–BiLSTM–CTC** recognizer. We additionally recognize text from an "ideal segmentation" method (100% segmentation accuracy) and compute its ideal-segmentation CER, which represents the best possible CER baseline for each recognition method. This allows us to quantify how close each segmentation method's downstream CER is to this baseline lower-bound under a *fixed recognition model*. We also use **Gemini-2.5-Flash for end-to-end transcription**, letting it recognize full-page text without external text-line coordinates.

**Computational Cost.**   We measure the per-page inference latency and the peak GPU memory (VRAM) usage of the competing methods when processing the full set of 481 images from the Sanskrit Dataset, whose resolutions are approximately $2500 \times 1450$ pixels. All methods were evaluated without batching (batch size of 1) on a workstation with an Intel Xeon w5-2465X CPU (32 cores), NVIDIA RTX A4500 GPU (20 GB), and 250 GB RAM. We utilized PyTorch profiling with GPU synchronization, and skipped the first warm-up page image to obtain stable measurements.

## 6   RESULTS

**Zero Shot Learning.**   In data-scarce settings, such as text-line segmentation, where annotation is time-consuming, the ability to generalize from limited data is highly valuable. With this in mind, we first evaluate if the proposed method, when trained purely on graph-based synthetic layout data, satisfactorily performs zero-shot text-line segmentation (as defined in Section 5) on real data. The results presented in Table 3 show that the proposed method achieves competitive performance on simple page layouts and superior performance on complex page layouts when trained solely on synthetic layout data. Table 3 also highlights the value added (measured by the higher AP@0.5 score) by the GNN-based methods (Branch 2 in Figure 1) over the DBSCAN-based anomaly detection method (Branch 1 in Figure 1).

Table 3: Zero-shot performance of proposed method on Sanskrit Dataset, compared with out-of-the-box pre-trained models Doc-UFCN and SeamFormer.

| Method | Simple Layouts | | | Complex Layouts | | |
|---|---|---|---|---|---|---|
| | AP(@0.5, @0.75) | IoU | F1 | AP(@0.5, @0.75) | IoU | F1 |
| Doc-UFCN | (0.85, 0.32) | 0.68 | 0.81 | (0.71, 0.12) | 0.57 | 0.72 |
| SeamFormer | (**0.96**, 0.19) | 0.66 | 0.79 | (0.76, 0.12) | 0.61 | 0.75 |
| Proposed Method (SplineCNN) | (0.92, **0.70**) | **0.80** | **0.88** | (**0.80, 0.46**) | **0.77** | **0.87** |
| Proposed Method (MPNN) | (0.91, 0.68) | **0.80** | **0.88** | (0.79, **0.46**) | **0.77** | **0.87** |
| Proposed Method (GAT) | (0.76, 0.56) | **0.80** | **0.88** | (0.73, 0.40) | 0.75 | 0.85 |
| Proposed Method (GCN) | (0.71, 0.51) | 0.78 | **0.88** | (0.69, 0.38) | 0.74 | 0.85 |
| Proposed Method (DBSCAN) | (0.64, 0.47) | **0.80** | **0.88** | (0.59, 0.34) | 0.76 | 0.86 |

**Distributional Robustness.**   We check the robustness of competing methods to distribution shifts with increasing data sizes (using inter-manuscript train-test data splits and intra-manuscript train-test data splits) as defined in Section 5. The results of the intra-manuscript train-test split experiment, as seen in Figure 2a, suggest that the Vision Transformer-based method SeamFormer is more *data-hungry* in comparison to the U-Net-based Doc-UFCN and the proposed method. When the training

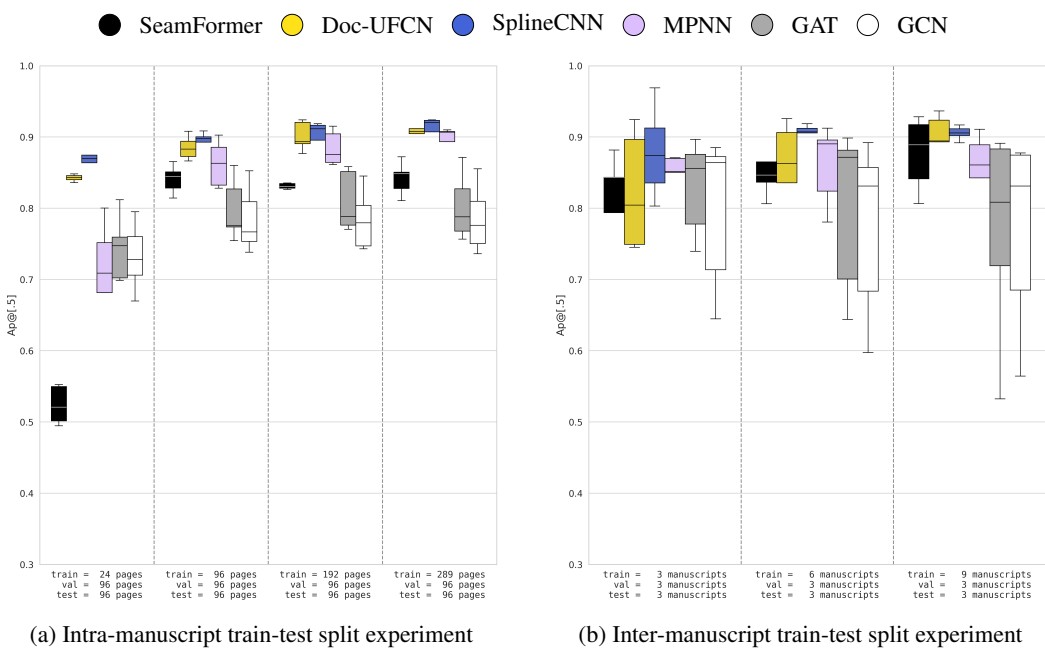

(a) Intra-manuscript train-test split experiment          (b) Inter-manuscript train-test split experiment

Figure 2: **Distributional Robustness.**    Performance comparison between competing methods with increasing training data sizes (X-axis), and with the resulting box plots showing the distribution of AP@0.5 values on the Y-axis with 5-fold data splits.

data size is small (24 pages), as seen in Figure 2a, the proposed method (GNN-SplineCNN) is observed to perform competitively in terms of consistency, and better in terms of accuracy. However, with increasing training dataset sizes, the gap in accuracy between the competing methods decreases. In the inter-manuscript train-test split experiment results seen in Figure 2b, we observe the proposed method (GNN-SplineCNN) to perform better in terms of accuracy and consistency than Doc-UFCN and SeamFormer. However, with increasing training data, Doc-UFCN grows more consistent and catches up with the proposed method's accuracy. Further results of this experiment using the misleading evaluation metrics IoU and F1 score (as discussed in Section 5 and Appendix B.1), and are shown in Figure 5.

**Ablation Study.**    The results of the ablation study defined in Section 5 demonstrate that GCN and GAT rely almost entirely on the one-hot edge overlap feature obtained from the heuristic algorithm. As shown in Table 6, when this feature is removed, the AP@0.50 score of GCN and GAT crashes from 0.73 and 0.78 down to 0.09 and 0.11, respectively. In contrast, without the one-hot edge overlap feature, MPNN and SplineCNN maintain high accuracy, dropping only slightly from 0.87 down to 0.82 and 0.84, respectively. This suggests that MPNN and SplineCNN can neurally and *implicitly* learn structural rules of text-lines without requiring *explicit* one-hot heuristic edge features.

**Downstream Evaluation.**    In evaluating the downstream text-recognition experiment defined in Section 5, we observe in Table 4 that all competing segmentation methods get CER scores close to the ideal segmentation CER on simple page layouts. However, on complex page layouts, the proposed method CER is closest to the ideal segmentation CER, being inferior by **3%**, Doc-UFCN being inferior by **16%**, and SeamFormer by **22%** when the recognition model is *fixed* as the finetuned CNN-BiLSTM-CTC. We observe similar percentages with the recognition models Pretrained CNN-BiLSTM-CTC (**4%, 13%, 10%**), and Layout-Guided Gemini-2.5-Flash (**5%, 8%, 13%**) in the same order. The traditional OCR pipeline—using the proposed method for text-line segmentation and a finetuned CNN-BiLSTM-CTC to recognize text from the resulting text-lines—demonstrates the best CER.

**Computational Cost.**    When comparing the computational costs of the competing methods under experiment settings defined in Section 5, we observe Doc-UFCN to be the most efficient, requiring only $0.053 \pm 0.013$ seconds per page and 262.7 MB of peak VRAM. The proposed GNN Pipeline

Table 4: Downstream page-level Character Error Rate (CER) comparison. A recognition method's CER (under Ideal Segmentation) corresponds to the best achievable CER for that recognition method when provided with perfectly segmented text-lines, and therefore reflects the limitations of the recognition method.

| Recognition Method (Segmentation Method) | CER (Simple Layout) | CER (Complex Layout) |
|---|---|---|
| Gemini Layout-Guided (SeamFormer) | 0.37 | 0.42 |
| Gemini Layout-Guided (Doc-UFCN) | **0.36** | 0.40 |
| Gemini Layout-Guided (Proposed Method) | **0.36** | **0.39** |
| Gemini Layout-Guided (Ideal Segmentation) | 0.36 | 0.37 |
| Pretrained CNN-BiLSTM-CTC (SeamFormer) | **0.38** | 0.51 |
| Pretrained CNN-BiLSTM-CTC (Doc-UFCN) | 0.43 | 0.52 |
| Pretrained CNN-BiLSTM-CTC (Proposed Method) | **0.38** | **0.48** |
| Pretrained CNN-BiLSTM-CTC (Ideal Segmentation) | 0.38 | 0.46 |
| Finetuned CNN-BiLSTM-CTC (SeamFormer) | 0.12 | 0.38 |
| Finetuned CNN-BiLSTM-CTC (Doc-UFCN) | 0.17 | 0.36 |
| Finetuned CNN-BiLSTM-CTC (Proposed Method) | **0.11** | **0.32** |
| Finetuned CNN-BiLSTM-CTC (Ideal Segmentation) | 0.11 | 0.31 |
| Gemini (End-to-End) | 0.39 | 0.44 |

(SplineCNN) is the second fastest, processing images at $1.588 \pm 0.608$ seconds per page, and incurs the highest VRAM memory footprint (3337.5 MB), which is driven by CRAFT and depends on the resolution of input images. Finally, SeamFormer is the slowest method ($11.156 \pm 3.093$ s/page), having moderate VRAM usage (519.8 MB).

## 7 Limitations and Future Work

As the foundation of the proposed method is CRAFT, documents that the pre-trained CRAFT model fails to detect characters from (e.g., pages from DRoSB Dataset (Fizaine et al., 2024)) cannot be processed by the proposed method. For such documents, CRAFT would need to be specifically fine-tuned in a supervised or semi-supervised setting, or adapted using test-time adaptation (TTA). It thus remains a topic of investigation to rigorously check if training a generalizable character segmenter is *inherently easier* than training a text-line segmenter, and if the proposed problem decomposition **(character segmentation+GNNs)** is a better representation of the problem than **end-to-end text-line segmentation**.

Although practical, the proposed method had significantly higher training time and per-page inference latency than Doc-UFCN, a production-grade tool, highlighting the need for further optimization. In future work, we aim to study GNN architecture design choices to better align with the problem formulation while optimizing latency, training efficiency and accuracy, and also aim to perform multi-task learning where the GNN identifies text-lines, text-boxes, and reading order.

The results in this preliminary study show encouraging evidence in support of the proposed graph-based problem formulation, which makes use of geometric priors of text-lines, allowing training on large-scale synthetic data simulating complex layouts. In a broader view, with the rapid advancement of Multimodal LLMs (MLLMs) demonstrating strong performance in digitizing document text end-to-end, we envision the future iterations of the proposed method serving as a visual grounding component for MLLMs (Chen et al., 2023; Wei et al., 2025; Chen et al., 2021), shifting away from the traditional pipeline of text-line segmentation followed by text-line recognition (Kiessling, 2025).

### Acknowledgments

The authors wish to express their thanks to Lalchand Research Library, DAV College, Chandigarh, India, for making manuscript data publicly available for educational and research purposes. The authors also wish to express their gratitude to the anonymous reviewers, Dr. Petar Veličković, Dr. Dhaval Patel, Dr. Tarinee Awasthi, Shagun Dwivedi, Ansh Kushwaha, and Dr. Oliver Hellwig for their invaluable support and feedback.

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

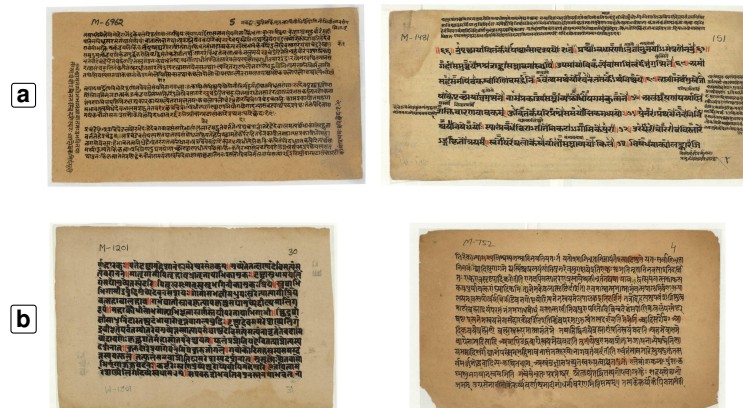

Figure 3: **Illustration of dataset pages with (a) Complex Layouts, (b) Simple Layouts.** Courtesy of Lalchand Research Library, DAV College, Sector 10, Chandigarh, India

Peng Zhang, Can Li, Liang Qiao, Zhanzhan Cheng, Shiliang Pu, Yi Niu, and Fei Wu. Vsr: a unified framework for document layout analysis combining vision, semantics and relations. In *International conference on document analysis and recognition*, pp. 115–130. Springer, 2021.

Shi-Xue Zhang, Xiaobin Zhu, Jie-Bo Hou, Chang Liu, Chun Yang, Hongfa Wang, and Xu-Cheng Yin. Deep relational reasoning graph network for arbitrary shape text detection. In *Proceedings of the IEEE/CVF conference on computer vision and pattern recognition*, pp. 9699–9708, 2020.

Shiyu Zhang, Caiying Zhou, Yonggang Li, Xianchao Zhang, Lihua Ye, and Yuanwang Wei. Irregular scene text detection based on a graph convolutional network. *Sensors*, 23(3):1070, 2023.

Zhenrong Zhang, Jiefeng Ma, Jun Du, Licheng Wang, and Jianshu Zhang. Multimodal pre-training based on graph attention network for document understanding. *IEEE Transactions on Multimedia*, 25:6743–6755, 2022.

Silvia Zottin, Axel De Nardin, Giuseppe Branca, Claudio Piciarelli, and Gian Luca Foresti. Icdar 2025 competition on few-shot text line segmentation of ancient handwritten documents (fest). In *International Conference on Document Analysis and Recognition*, pp. 586–602. Springer, 2025.

## A    LAYOUT TYPES IN SANSKRIT MANUSCRIPTS

Pages of Sanskrit Manuscripts with Complex Layouts and Simple Layouts are illustrated in Figure 3.

## B    LABEL BIAS

To standardize the dataset and enable training and comparison with methods that require bounding polygon labels, such as Doc-UFCN (Boillet et al., 2021) and SeamFormer (Vadlamudi et al., 2023), there is a need to convert graph-based labels, as seen in Figure 4(a), to bounding polygons, Figure 4(b). To do this, we binarize the CRAFT heatmap using a threshold of 130. The areas of the images that pass the threshold are then enclosed in bounding boxes. These bounding boxes, sometimes incorrectly, contain diacritic marks from adjacent lines. To exclude these, we use a dynamic cropping technique that crops the bounding boxes at their top or bottom if diacritic marks are detected touching the top side or bottom side of the box. Next, the disconnected bounding boxes that belong to the same logical line are consolidated into a single contour using an iterative merging process. In this step, all boxes for a line are first rasterized into a mask, and any resulting disconnected regions are repeatedly linked by drawing rectangular "bridges" between the closest fragments until

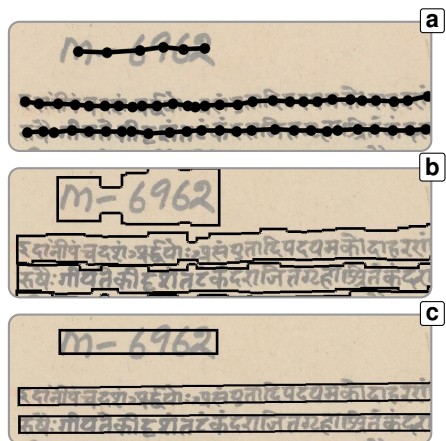

Figure 4: **Label Conversion and Normalization.** (a) Graph-based ground truth labels. (b) Loose bounding polygonal ground truth labels. (c) Non-overlapping simplified labels.

only one connected component remains. The outer contour of this unified region is then extracted to obtain the final polygon, as shown in Figure 4(b).

These bounding polygons can train SeamFormer (Vadlamudi et al., 2023), but because they sometimes overlap with polygons from adjacent lines, they cannot be used to train Doc-UFCN, which *necessarily requires non-overlapping ground-truth polygons*. Hence, we further simplify the ground truth by converting bounding polygons to non-overlapping simplified bounding polygons as seen in Figure 4(c).

As investigated by Boillet et al. (2022), methods trained on one set of annotations underperform on another set of ground truth annotations with a different annotation style. We minimize this label bias by training and evaluating Doc-UFCN on the simplified non-overlapping ground truth bounding polygons it necessarily requires. As SeamFormer predicts precise tight polygons around the text-lines, its predictions are also better suited to be compared with the simplified bounding rectangles than the bounding polygons. However, this means that the labels are slightly biased against SeamFormer. The labels are also slightly biased in favour of the proposed method, because the same post-processing is applied to both the predicted graph and the ground-truth graph to obtain bounding polygons. Due to this reason, we primarily focus on reporting results for the evaluation metric **AP@0.5** (as recommended by Boillet et al. (2022)), which is minimally affected by label bias. This lenient threshold of 0.5 is accommodating of *stylistic differences* between types of text-line label annotations, ensuring a fair comparison with SeamFormer.

Specifically, pixel-level metrics such as IoU and F1 are overly sensitive to annotation style, often penalizing valid predictions solely for stylistic differences—such as margin width or contour complexity—relative to the Ground Truth. Since the Proposed Method shares the exact same polygon-generation logic as the Ground Truth, it benefits from an inherent geometric conformity that inflates its pixel-level metric performance. Conversely, SeamFormer predicts tightly-fitted contours that may semantically capture the text perfectly but diverge stylistically from the simplified non-overlapping ground truth bounding polygons. By utilizing AP@0.5, we decouple the evaluation of *localization* (finding the text-line) from *bounding polygon style* (mimicking the specific annotation style), preventing the Proposed Method from receiving an unfair advantage based solely on the algorithmic construction of the bounding polygons.

To summarize, GNN-based methods are trained on the graph-based labels seen in Figure 4 (a), SeamFormer is trained on the bounding polygon labels in Figure 4 (b), and Doc-UFCN is trained using the non-overlapping simplified bounding polygons in Figure 4 (c). The predictions of all three methods are evaluated with the ground truth simplified rectangular bounding polygons seen in Figure 4 (c).

### B.1 Distributional Robustness (Sanskrit Dataset, IoU and F1 Evaluation Metrics)

As studied by Boillet et al. (2022), IoU and F1 score metrics show misleading results in the Distributional Robustness experiment setting, on the Sanskrit Dataset, in Figure 5 due to label bias, by unfairly favoring the proposed method, which has been used to annotate the ground-truth labels. In other words, the stylistic differences between possible types of text-line label annotations cause unfairly higher IoU and F1 scores for the method used to annotate the ground-truth text-line annotations.

Essentially, since the ground-truth boundaries share the exact geometric logic (such as padding or polygon tightness) as the proposed method, the metric becomes a measure of stylistic conformity rather than detection accuracy. *Competing methods are therefore penalized simply for drawing the bounding box differently, even if they have correctly detected the text-line location.*

Due to this reason, we primarily focus on reporting results for the evaluation metric **AP@0.5** (as implemented by Boillet et al. (2022)), which is minimally affected by label bias.

## C DivaHisDB Dataset

On the standard dataset DIVA-HisDB (Simistira et al., 2016), we compare the data scaling performance of the competing methods using the intra-manuscript train–test split experimental setting. An inter-manuscript train–test data split experiment was not feasible due to the dataset containing just 3 sub-dataset documents. For this dataset, as text-line labels are in the form of tight bounding polygons, we perform Label normalization (while accounting for CRAFT failures) to convert the tight polygons to graph-based labels and rectangular labels to mitigate the label bias.

We show the results in Figure 6 for the metric AP@0.5, IoU, and F1 score. Similar to the Sanskrit Dataset results, these results suggest that the Vision Transformer-based method SeamFormer is more *data-hungry* in comparison to the U-Net-based Doc-UFCN and the proposed method. When the training dataset size is as low as 6 pages, as seen in Figure 6a, the proposed method (GNN-SplineCNN) is observed to perform competitively with Doc-UFCN accuracy, and better in terms of consistency, as indicated by their respective box plot lengths. However, with increasing training dataset sizes, the performance increase for the proposed method stalls, indicating possible errors made by pre-trained CRAFT in the character detection stage.

## D U-DIADS-TL Dataset

On the U-DIADS-TL (Zottin et al., 2025) dataset introduced at the FEST Competition 2025, we evaluate the performance of the proposed method in both zero-shot and few-shot settings, using the same intra-manuscript data splits prescribed by the competition. In the zero-shot setting, we trained a SplineCNN on 10K synthetic data points, simulating diverse layouts. In the few-shot setting, we convert the 3 pages of training data into the graph-based format, augment each page 500 times, and merge the augmented training data with the 10K synthetic dataset. To detect characters, the pre-trained CRAFT is used as-is. In order to compare with the performance of the FEST Competition 2025 participants, we do not perform label normalization and instead perform post-processing on the predicted polygons. To convert the loose bounding polygon predictions of the proposed method to precise pixel-level predictions, we had to implement a post-processing step, which overlayed the predicted loose polygons with binarized versions of the original images formed using Adaptive thresholding, DocEnTr (Quattrini et al., 2024), and the method by Souibgui et al. (2022). Finally, a baseline is dynamically drawn using a projection profile and Principal Component Analysis (PCA) approach to imitate the baselines in the precise ground truth annotations. We used the provided validation data to perform early stopping during model training and also to refine the post-processing step, which required primary effort, since the IoU threshold used to compute the object-level metric LineIU is a strict 0.75.

In Table 5, we observe the proposed method to perform competitively with leading methods on Latin2 and Latin1439 manuscripts from the U-DIADS-TL dataset introduced at the FEST Competition 2025. An interesting finding is that the performance of the zero-shot experiment was on par

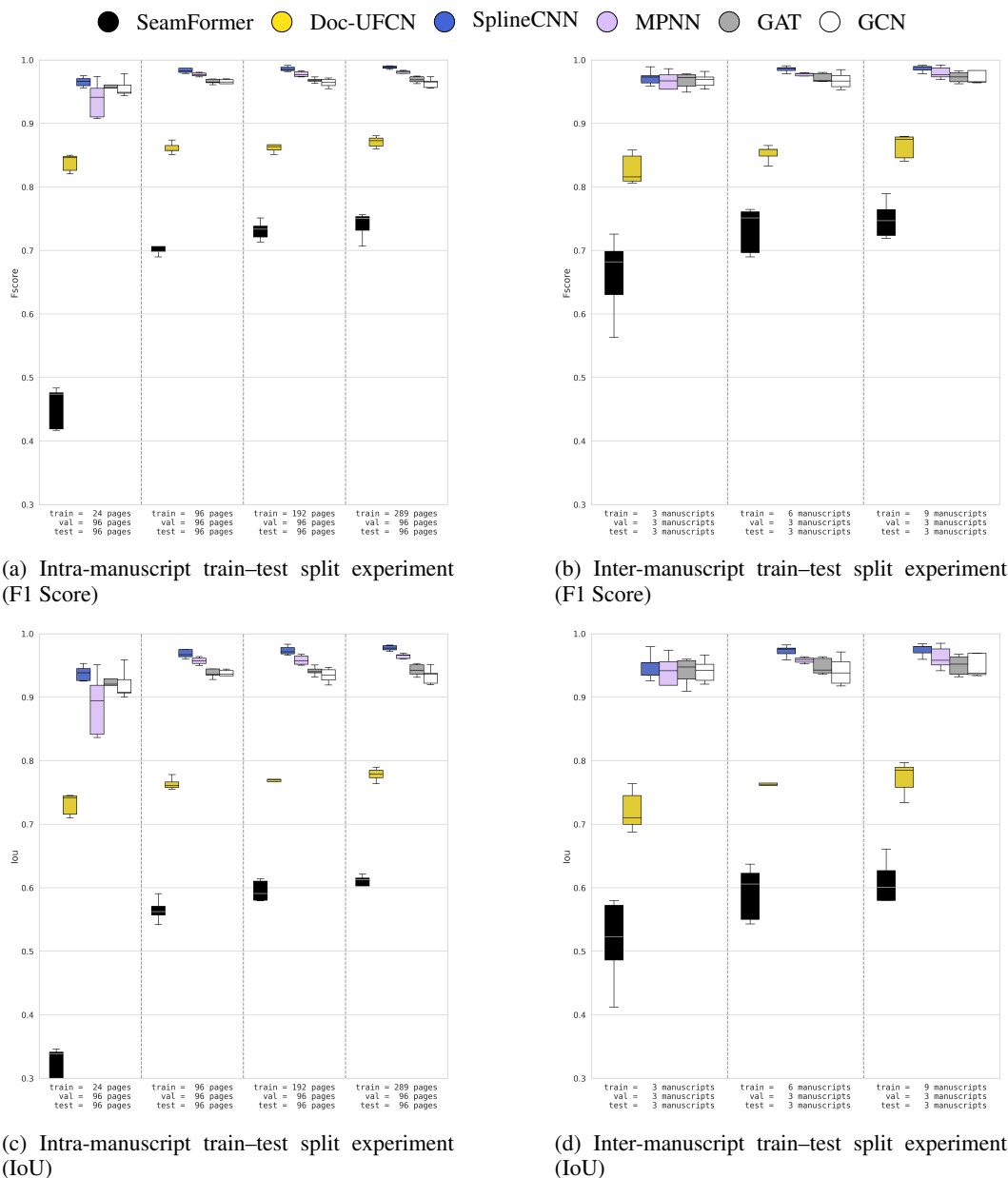

Figure 5: **Distributional Robustness (Sanskrit Dataset).** Performance comparison between competing methods under intra-manuscript train–test split and inter-manuscript train–test split experimental settings with increasing training data sizes (X-axis), with the resulting box plots showing the distribution of **IoU and F1 score** values on the Y-axis across 5-fold data splits.

with the few-shot experiment. However, performance on Syr341 was impacted negatively because of unsatisfactory character detection (by CRAFT) and binarization.

# E ABLATION STUDY

The heuristic algorithm described in Section 4 explicitly uses geometric priors of text-lines to generate discrete, one-hot node features (node degree) and edge features (edge overlap frequency) as shown by the node and edge colors in Figure 1 (c).

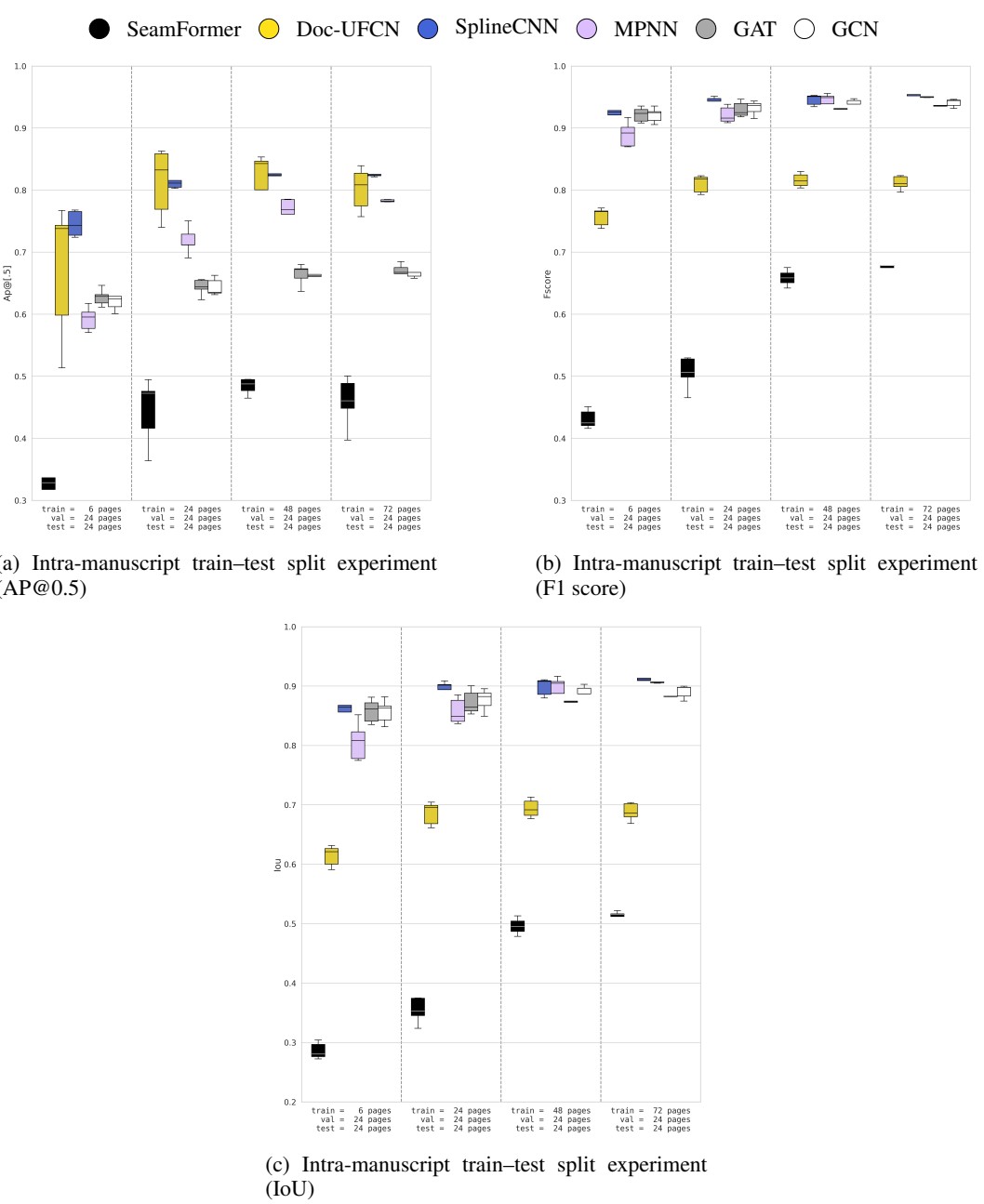

(a) Intra-manuscript train–test split experiment (AP@0.5)

(b) Intra-manuscript train–test split experiment (F1 score)

(c) Intra-manuscript train–test split experiment (IoU)

Figure 6: **Distributional Robustness (DivaHisDB Dataset).** Performance comparison between competing methods under the intra-manuscript train–test split experimental setting with increasing training data sizes (X-axis), with the resulting box plots showing the distribution of **AP@0.5, IoU, and F1** values on the Y-axis across 5-fold data splits.

From an algorithmic alignment perspective, rather than relying on explicit one-hot features obtained using the heuristic algorithm, a well-aligned GNN should neurally learn the structural rules of text-lines implicitly from the raw, continuous geometric features: the normalized 2D coordinates for nodes, and the horizontal offsets, vertical offsets, Euclidean distances, and aspect ratios for edges as defined in Table 2. To test this, we evaluate our four GNN architectures—GCN, GAT, MPNN, and SplineCNN—under five distinct feature ablations:

Table 5: Zero-shot and few-shot performance of the proposed method on dataset U-DIADS-TL, compared with leading FEST Competition 2025 participants

| Method | Latin2 | | | | | Latin14396 | | | | | Syr341 | | | | |
|---|---|---|---|---|---|---|---|---|---|---|---|---|---|---|---|
| | PIU | LIU | DR | RA | FM | PIU | LIU | DR | RA | FM | PIU | LIU | DR | RA | FM |
| PERO | 0.79 | **0.97** | 0.80 | **0.87** | **0.83** | 0.76 | **0.94** | 0.70 | 0.72 | 0.71 | **0.86** | **0.97** | 0.94 | **0.96** | **0.95** |
| CV-Group | **0.83** | 0.96 | **0.87** | 0.18 | 0.30 | 0.67 | 0.77 | 0.63 | 0.14 | 0.22 | 0.85 | 0.97 | **0.96** | 0.19 | 0.31 |
| SRCB | 0.78 | 0.94 | 0.85 | 0.78 | 0.81 | 0.75 | 0.90 | 0.75 | 0.73 | 0.74 | 0.81 | 0.94 | 0.92 | 0.88 | 0.90 |
| TAU-CH | 0.71 | 0.86 | 0.68 | 0.74 | 0.70 | 0.65 | 0.61 | 0.19 | 0.17 | 0.18 | 0.72 | 0.90 | 0.63 | 0.69 | 0.66 |
| GPI | 0.78 | 0.93 | 0.73 | 0.79 | 0.75 | 0.71 | 0.76 | 0.34 | 0.34 | 0.34 | 0.74 | 0.88 | 0.62 | 0.59 | 0.60 |
| VAI-OCR | 0.72 | 0.83 | 0.79 | 0.84 | 0.81 | 0.55 | 0.65 | 0.58 | 0.70 | 0.64 | 0.67 | 0.76 | 0.75 | 0.84 | 0.79 |
| GNN (Zero-Shot) | 0.80 | 0.82 | 0.78 | 0.81 | 0.79 | 0.85 | 0.92 | **0.92** | 0.91 | 0.91 | 0.74 | 0.65 | 0.53 | 0.53 | 0.53 |
| GNN (SplineCNN) | 0.80 | 0.85 | 0.80 | 0.83 | 0.81 | **0.85** | 0.92 | **0.92** | **0.92** | **0.92** | 0.74 | 0.72 | 0.56 | 0.56 | 0.56 |
| GNN (MPNN) | 0.80 | 0.86 | 0.81 | 0.83 | 0.82 | 0.85 | 0.92 | **0.92** | **0.92** | **0.92** | 0.74 | 0.71 | 0.56 | 0.56 | 0.56 |
| GNN (GAT) | 0.75 | 0.62 | 0.65 | 0.68 | 0.66 | 0.83 | 0.83 | 0.86 | 0.87 | 0.87 | 0.69 | 0.39 | 0.37 | 0.36 | 0.36 |
| GNN (GCN) | 0.76 | 0.58 | 0.62 | 0.63 | 0.62 | 0.81 | 0.79 | 0.83 | 0.85 | 0.84 | 0.71 | 0.36 | 0.38 | 0.33 | 0.35 |
| LDLD | 0.35 | 0.00 | 0.00 | 0.00 | 0.00 | 0.39 | 0.00 | 0.00 | 0.00 | 0.00 | 0.42 | 0.00 | 0.00 | 0.00 | 0.00 |
| Codecrackers | 0.52 | 0.13 | 0.03 | 0.02 | 0.02 | 0.53 | 0.47 | 0.23 | 0.18 | 0.20 | 0.26 | 0.01 | 0.00 | 0.00 | 0.00 |
| DIA-Group | 0.65 | 0.72 | 0.58 | 0.39 | 0.46 | 0.31 | 0.22 | 0.15 | 0.17 | 0.16 | 0.67 | 0.78 | 0.74 | 0.62 | 0.67 |
| CV-Lab | 0.73 | 0.80 | 0.78 | 0.44 | 0.56 | 0.58 | 0.79 | 0.59 | 0.42 | 0.49 | 0.74 | 0.79 | 0.79 | 0.55 | 0.64 |
| BBA | 0.52 | 0.26 | 0.23 | 0.12 | 0.16 | 0.47 | 0.09 | 0.05 | 0.03 | 0.04 | 0.56 | 0.31 | 0.30 | 0.15 | 0.20 |
| DeepLabV3+ | 0.55 | 0.53 | 0.28 | 0.15 | 0.19 | 0.57 | 0.58 | 0.51 | 0.33 | 0.40 | 0.18 | 0.08 | 0.03 | 0.04 | 0.03 |
| PSPNet | 0.52 | 0.40 | 0.15 | 0.11 | 0.13 | 0.50 | 0.52 | 0.34 | 0.30 | 0.32 | 0.08 | 0.02 | 0.00 | 0.00 | 0.00 |
| FCN | 0.51 | 0.45 | 0.29 | 0.17 | 0.21 | 0.55 | 0.49 | 0.44 | 0.27 | 0.33 | 0.34 | 0.22 | 0.09 | 0.05 | 0.06 |

1. **All Features:** The complete set of raw continuous geometric features and the one-hot features generated by the heuristic algorithm.

2. **No One-Hot Node:** Removal of the one-hot node degree feature.

3. **No One-Hot Edge:** Removal of the one-hot edge overlap feature.

4. **No One-Hot:** Removal of both the one-hot node degree and edge overlap features (relying strictly on raw continuous geometric features).

5. **Only One-Hot:** Retaining *only* the one-hot node and edge features, completely removing the raw continuous geometric features.

Table 6 details the resulting text-line segmentation performance (AP@0.50) on all 481 pages of the Sanskrit dataset, where all models were trained on the same synthetic data for a fair comparison.

Table 6: Ablation study on node and edge features. Performance is measured in AP@0.50. "One-Hot Node" and "One-Hot Edge" refer to the discrete one-hot features generated by the heuristic algorithm (node degree and edge overlap).

| Model | Graph Feature Ablations (AP@0.50 ↑) | | | | |
|---|---|---|---|---|---|
| | All Features | No One-Hot Node | No One-Hot Edge | No One-Hot | Only One-Hot |
| GCN | 0.73 | 0.71 | 0.09 | 0.10 | 0.35 |
| GAT | 0.78 | 0.74 | 0.11 | 0.33 | 0.52 |
| MPNN | **0.87** | 0.86 | 0.82 | 0.61 | **0.80** |
| SplineCNN | **0.87** | **0.88** | **0.84** | **0.84** | **0.80** |

## F  DOWNSTREAM EVALUATION

We evaluate the goal-oriented downstream task of recognizing the text content in historical page images using the **page-level Character Error Rate (CER)**. CER measures the quality of the predicted transcription relative to the ground-truth text and is computed as the Levenshtein distance divided by the total number of characters in the ground-truth text. To calculate CER at the page-level as prescribed in Boillet et al. (2022), we sort predicted and ground-truth text-line polygons of an image from top-left to bottom-right. Then, all transcriptions are concatenated in a single line of text

following this order to calculate the CER. To perform this evaluation, we manually annotated the ground-truth segmentation and transcription for two previously unseen Sanskrit manuscripts—one with a simple layout and one with a complex layout.

For both manuscripts, we use the pretrained competing methods Doc-UFCN, SeamFormer, and the proposed method (GNN-SplineCNN) to predict the text-line segments of each page. To predict the text content from the segmented text-lines, we use the following recognition models: (1) **Gemini-2.5-Flash**, by passing the predicted text-line coordinates as part of the prompt; (2) a **pretrained CNN–BiLSTM–CTC** recognizer; and (3) a **finetuned CNN–BiLSTM–CTC** recognizer. We also recognize text from an "ideal segmentation" method (which has 100% segmentation accuracy) and compute the ideal-segmentation CER—i.e., the best possible CER for a given recognition model. This allows us to quantify how close each competing segmentation method's downstream CER is to the best possible CER for a recognition model, *which is kept fixed*.

To perform recognition by keeping the recognition model Gemini-2.5-Flash fixed, we use the prompt shown in Section F.2, where we pass the detected text-line locations as path traces in the prompt. These path traces are normalized to an integer range of 0–1000 following the conventions introduced by Chen et al. (2021) and Chen et al. (2023), using which Gemini has been trained to understand spatial coordinates. For the proposed method, the graph corresponding to each text-line is used as the path trace. As Doc-UFCN and SeamFormer predict bounding polygons rather than explicit baselines, we derive the path trace by estimating the centerline of each polygon to generate a smooth trajectory that faithfully follows the handwritten text flow.

In traditional OCR pipelines, text-line images are required to be passed to a CNN-BiLSTM-CTC based recognizer, which recognizes the text content from the text-line images. Hence, we first prepare the text-line images by cropping each method's predicted text-line polygons, applying appropriate padding, and rendering the extracted regions onto a background initialized with the page's median color before passing them to the recognizer. This CNN-BiLSTM-CTC recognition model may be either pretrained or finetuned (to perform appearance-level domain adaptation on historically styled text-lines).

We also prompt Gemini-2.5-Flash to perform end-to-end transcription without any external layout guidance using the prompt in Section F.1. This allows us to quantify Gemini's performance improvement when text-line path traces are provided externally compared to when they are not.

## F.1    TRANSCRIPTION USING GEMINI END-TO-END

```
prompt_text = """
You are an expert Indologist and Paleographer specializing in
    handwritten Sanskrit manuscripts.
Your Task: Perform a diplomatic transcription (OCR) of the manuscript
    image and provide bounding boxes.

CRITICAL INSTRUCTIONS:
1. Output Format: Output ONLY raw valid JSON. No Markdown.
2. Coordinates: You MUST provide bounding boxes for every text line and
    region.
   - Format: [ymin, xmin, ymax, xmax]
   - Scale: Normalized coordinates from 0 to 1000 (where 1000 is the
   full width/height).
3. Granularity: Transcribe at the VISUAL TEXT-LINE level.
4. Script: Unicode Devanagari.

JSON SCHEMA:
{
  "status": "success",
  "regions": [
    {
      "id": "region_0",
      "type": "main_text",
      "box_2d": [ymin, xmin, ymax, xmax],
      "lines": [
        {
```

```
            "id": "line_0",
            "box_2d": [ymin, xmin, ymax, xmax],
            "text": "Transcribed text here"
          }
        ]
      }
    ]
}
"""
```

## F.2   TRANSCRIPTION USING LAYOUT-GUIDED GEMINI

```
prompt_text = (
    "You are an expert Indologist and Paleographer specializing in
    handwritten Sanskrit manuscripts."
    "Your Task: Perform a diplomatic transcription (OCR) of the attached
    manuscript image.\n"
    "CRITICAL INSTRUCTIONS:\n"
    "Transcribe the Sanskrit text from the image at the text-line level,
    where locations of the handwritten text-lines are defined using
    'Path Traces'. Each 'Path Trace' refers to one text-line.\n"
    "The coordinates of the Path Traces are normalized on a 0-1000 scale
    (where [0,0] is top-left and [1000,1000] is bottom-right) "
    "to precisely map the text line locations on the image.\n"
    "For each path trace [y_start, x_start, y_mid1, x_mid1, y_mid2,
    x_mid2, y_end, x_end], transcribe the text that sits along this
    curve.\n"
    "Focus strictly on the visual line indicated by the trace; ignore
    text from lines above or below.\n"
    "Transcribe in Unicode Devanagari. Preserve original spelling
    (Sandhi).\n"
    "Output a JSON array of objects with 'id' and 'text'.\n\n"
    "REGIONS:\n"
)
for item in regions_payload:
    prompt_text += f"ID: {item['id']} | Trace: {item['trace']}\n"
```

