# OpenReview forum: "Towards Text-Line Segmentation of Historical Documents Using Graph Neural Networks"
_ICLR.cc/2026/Workshop/GRaM — ICLR 2026 Workshop GRaM Poster_

### Official Review · Reviewer_aMGJ · 2026-02-08
**Well-executed graph-based document segmentation with missing ablations**

**Rating:** 7
**Confidence:** 3

**Review:**

The paper formulates text-line segmentation of historical manuscripts as a graph problem: CRAFT detects characters (nodes), a heuristic algorithm connects adjacent characters using geometric priors (angle ~180 degrees, character spacing < line spacing), and a GNN performs binary edge classification to refine connections. The authors introduce a 15-manuscript Sanskrit dataset (481 pages) and show competitive results against Doc-UFCN and SeamFormer in zero-shot, in-distribution, and out-of-distribution settings.

**What works well:**

1. The problem decomposition is well-motivated. Separating character detection (appearance-level, handled by CRAFT) from line grouping (layout-level, handled by GNN on geometric features) has a clear advantage: the GNN can be trained on synthetic layout data, making it robust to layout distribution shifts without needing real annotated pages. The zero-shot results in Table 3 support this. The GNN trained on 50K synthetic data points achieves AP@0.5 of 0.80 on complex Sanskrit layouts versus 0.71 for Doc-UFCN and 0.76 for SeamFormer, both pretrained models.

2. Strong fit for GRaM. This is exactly the kind of work the workshop targets: geometric inductive priors like angle constraints and distance ordering encoded into a graph structure, processed by GNNs. The node and edge features in Table 2 are all geometric: normalized position, degree, relative displacement, Euclidean distance, aspect ratio.

3. The evaluation methodology is careful. Using AP@0.5 instead of IoU/F1 to avoid label bias, as advocated by Boillet et al. (2022), is the right call, and the paper explains why in detail in Appendix A.2. The 5-fold cross-validation with box plots, fixed validation/test splits, and nested training set sizes make results interpretable. The downstream OCR evaluation in Table 4 using three different recognizers is a practical validation.

4. The synthetic data pipeline is a genuine contribution. Rejection sampling for diverse layouts, micro-jittering for handwriting variation, textbox splitting for structural ambiguity: these design choices are grounded in real challenges of historical manuscript analysis.

5. The FEST Competition 2025 results in Table 5 provide external validation. The proposed method performs competitively with top participants like PERO and SRCB on Latin manuscripts, and the zero-shot performance matches few-shot on two of three test sets.

**Problems:**

1. CRAFT is a single point of failure and its limitations are not quantified. The paper acknowledges this in the discussion section but never reports CRAFT's character detection recall or precision on the Sanskrit dataset. If CRAFT misses 10% of characters, the GNN cannot recover those connections. The poor Syr341 results in Table 5 are attributed to "unsatisfactory character detection" by CRAFT, but no numbers are given. This is the most important ablation missing from the paper.

2. The baseline comparison for the GNN's added value over the heuristic graph is missing. Branch 1 (heuristic graph + DBSCAN anomaly detection + human correction) versus Branch 2 (heuristic graph + GNN edge classification): how much does the GNN actually improve over DBSCAN? The paper never isolates this comparison. Without it, we cannot assess whether the GNN contribution is marginal or substantial.

3. The SplineCNN vs MPNN vs GAT vs GCN comparison in Figures 2 and 5 shows SplineCNN and MPNN consistently outperform GAT and GCN. The paper attributes this to edge feature usage in layer-wise updates. But there is no ablation removing edge features from SplineCNN/MPNN to confirm this explanation. This would be a simple and informative experiment.

4. The geometric priors (K=10 nearest neighbors, 180+/-40 degree angle constraint) are tuned for roughly horizontal text. The paper does not discuss how these would adapt to vertical scripts like Chinese, Japanese, or Mongolian, nor to circular text or severely warped pages. For a GRaM paper, discussing the generality of the geometric assumptions would strengthen the contribution.

5. The label bias issue in Appendix A.2 is handled transparently but creates an uncomfortable situation: the proposed method's bounding polygon generation shares the same logic as the ground truth. The paper correctly uses AP@0.5 to mitigate this, but the IoU/F1 numbers in Figures 5-6 are explicitly called out as "misleading" by the authors themselves. This should be stated more prominently in the main text, not buried in the appendix.

**Missing references:**

- Renton et al. (2018), "Fully Convolutional Network with Dilated Convolutions for Handwritten Text Line Segmentation," an early FCN approach for this task that is worth citing.
- The paper does not discuss LayoutLM (Xu et al., 2020) or DocTR families, which use transformer-based document understanding with spatial features. While different in approach, they represent the current direction in document AI and the relationship should be acknowledged.

**Minor:**

- The heuristic graph algorithm description in Section 4 is dense. A pseudocode listing would improve clarity.
- Table 4 downstream CER numbers: on complex layouts, proposed method gets 0.32 CER vs 0.31 ideal (finetuned CNN-BiLSTM-CTC). That is only 1% gap, which is impressive, but the absolute CER of 0.32 means one in three characters is wrong. Worth noting this reflects the recognizer's limitations, not the segmentation.

**Questions:**

1. What is CRAFT's character detection recall on the Sanskrit dataset? How many ground-truth graph edges are missing from the heuristic graph?
2. How does Branch 1 (DBSCAN) compare to Branch 2 (GNN) on AP@0.5, without human correction?
3. The 180+/-40 degree constraint: how sensitive are results to this parameter? Have you tried 180+/-30 or 180+/-50?

**Assessment:** A well-executed workshop paper with natural alignment to GRaM's theme. The graph formulation, synthetic data pipeline, and new dataset are solid contributions. The main gap is the missing CRAFT quality analysis and heuristic-vs-GNN ablation. The evaluation is thorough and honest about its limitations regarding label bias. Recommended for acceptance.

**Pmlr Suitability:**

Yes

---

### Official Review · Reviewer_fNrw · 2026-02-16
**An original graph-based approach to text-line segmentation in historical manuscripts**

**Rating:** 7
**Confidence:** 3

**Review:**

## Strengths

- **Original problem and proposed approach**: The paper proposes an original graph-based approach to text-line segmentation. By framing the task as binary edge classification between character nodes, it offers a fresh alternative to traditional pixel-level methods.

- **Relevant contextualization**: The literature review is thorough and effectively places the paper’s contributions within the current research landscape, making the problem and the approach clear.

- **New benchmark and data pipeline**: The introduction of a new Sanskrit manuscript dataset seems to be a significant contribution to the field. In addition, the ability to generate large-scale synthetic data for training provides an advantage for data augmentation reducing the need for manual annotation.

- **Clarity of methodology**: The method is described with clarity. The inclusion of figures and tables (such as the node and edge feature summary) provides a good visual support for better understanding the pipeline.

- **Comprehensive evaluation beyond the proposed benchmark**: The authors go beyond their own benchmark (in the appendix) by evaluating the model on existing datasets like U-DIADS-TL and DIVA-HisDB, proving the method is adaptable to different settings (although it would be interesting to include some of these results in the main text).

## Weaknesses

- **Computational cost**: The multi-stage pipeline involving CRAFT detection, heuristic graph construction, and GNN inference appears computationally expensive. In Section 7 (Limitations), the authors admit that Doc-UFCN is the fastest in terms of training and inference. However, they do not provide a formal complexity analysis of the method. A detailed complexity analysis and a direct runtime comparison with the end-to-end baselines (Doc-UFCN and SeamFormer) would be important for fairness.

- **Baseline fairness**: In Section 5.1 (Zero Shot), the authors state they compare their method against "publicly available pre-trained models" for Doc-UFCN and SeamFormer. Since the proposed GNN was trained on 50k synthetic samples while the baselines were used in their pre-trained version, comparing the proposed model to "out-of-the-box" pre-trained baselines may not be entirely fair. For fairness, the baselines could have been fine-tuned as well. Specifically, if one fine-tunes Doc-UFCN on the same synthetic layouts, would it perform just as well?

- **Ablations**:  While authors compare different GNN layers (GCN vs. GAT vs. MPNN), the paper would benefit from a dedicated ablation study to isolate the impact of specific components, such as the geometric priors, the $K$-NN connectivity, or the specific choice of GNN architecture.

- (Minor) **Conclusion**: While the "Limitations and Discussion" section is appreciated, the paper could further be strengthened with a conclusion.

- (Minor) Some citations are inconsistent (e.g. Line 39 misses the year in the citation).

**Pmlr Suitability:**

Yes

---

### Official Review · Reviewer_1VwC · 2026-02-23
**Smart 'Geometric Engineering' in text-line segmentation using geometric priors and graph networks**

**Rating:** 6
**Confidence:** 5

**Review:**

## Paper Insights
The authors have proposed a novel idea of reducing the problem of text segmentation to a binary edge classification problem, which in itself is a very interesting problem casting. Through this approach it is interesting to see how the authors have managed to decouple the problem and provide a really interesting solution. The focus on zero shot generalization is a smart application case, with modest theoretical contributions.

The paper is very well written and the ideas have been clearly expressed with sound explanations of each architectural choice. It is very easy to follow and understand the intuition, motivation of the authors and their design choices. The simplicity of the geometric prior being such a strong inductive bias, and recomposing the problem from this lens, is exactly what makes it a strong fit for the workshop.

## Suggestions

Despite it being a strong fit for the workshop I would take this opportunity to mention that the paper lacks strong theoretical contribution and seems a bit more 'geometric engineering'. The geometric prior although interesting is a bit trivial, and can be improved by making it learnable instead of algorithmic (even through supevision, since the authors create a synthetic dataset). Further, the edge features during message passing are easily ignored and there isnt much discussion on trying to resolve it. This is a significant missed opportunity since the geometric edge features are arguably the most informative signals. Fully edge-conditioned message passing, would be interesting to try out and study. The paper gestures at this with SplineCNN and MPNN performing better than GAT/GCN, but doesn't fully exploit it.

I would be interested in hearing the authors' opinions on the same.

## Final Thoughts
I believe this is a good fit for the workshop but could help with some improvements in design choices to make it a more theoretical contribution than an engineering design choice. It could help with some more ablation studies show the impact of specific components, such as the geometric priors, and the specific choice of GNN architecture.

**Pmlr Suitability:**

No

---

### Meta-Review · Area_Chair_ZVAg · 2026-02-28

**Decision:**

Accept

**Metareview:**

I agree with the reviewers and suggest additional evidence theoretical or ablation experiments to showcase the effectiveness of the proposed method.

I recommend an accept but request the authors to take some time to address the concerns and suggestions by the reviewers and update the draft for the camera ready version.

**Relevance To Proceedings:**

Yes — suitable for PMLR (long paper)

**Relevance To Workshop:**

Yes — suitable for GRaM

---

### Decision · Program_Chairs · 2026-03-02

Accept (Poster)